# Signatures of Co-Deregulated Genes and Their Transcriptional Regulators in Lung Cancer

**DOI:** 10.3390/ijms231810933

**Published:** 2022-09-18

**Authors:** Angeliki Chatziantoniou, Apostolos Zaravinos

**Affiliations:** 1Department of Life Sciences, School of Sciences, European University Cyprus, Nicosia 1516, Cyprus; 2Cancer Genetics, Genomics and Systems Biology Laboratory, Basic and Translational Cancer Research Center (BTCRC), Nicosia 1516, Cyprus

**Keywords:** lung cancer, tumor heterogeneity, Characteristic Direction, co-deregulated genes, single-gene perturbation, single-drug perturbation, drug repurposing, GEO2Enrichr, X2K

## Abstract

Despite the significant progress made towards comprehending the deregulated signatures in lung cancer, these vary from study to study. We reanalyzed 25 studies from the Gene Expression Omnibus (GEO) to detect and annotate co-deregulated signatures in lung cancer and in single-gene or single-drug perturbation experiments. We aimed to decipher the networks that these co-deregulated genes (co-DEGs) form along with their upstream regulators. Differential expression and upstream regulators were computed using Characteristic Direction and Systems Biology tools, including GEO2Enrichr and X2K. Co-deregulated gene expression profiles were further validated across different molecular and immune subtypes in lung adenocarcinoma (TCGA-LUAD) and lung adenocarcinoma (TCGA-LUSC) datasets, as well as using immunohistochemistry data from the Human Protein Atlas, before being subjected to subsequent GO and KEGG enrichment analysis. The functional alterations of the co-upregulated genes in lung cancer were mostly related to immune response regulating the cell surface signaling pathway, in contrast to the co-downregulated genes, which were related to S-nitrosylation. Networks of hub proteins across the co-DEGs consisted of overlapping TFs (SOX2, MYC, KAT2A) and kinases (MAPK14, CSNK2A1 and CDKs). Furthermore, using Connectivity Map we highlighted putative repurposing drugs, including valproic acid, betonicine and astemizole. Similarly, we analyzed the co-DEG signatures in single-gene and single-drug perturbation experiments in lung cancer cell lines. In summary, we identified critical co-DEGs in lung cancer providing an innovative framework for their potential use in developing personalized therapeutic strategies.

## 1. Introduction

Lung cancer (LC) maintains the highest mortality rate among cancer-related deaths, with almost an equal distribution between females and males [1]. Molecularly diverse subtypes of lung cancer have been investigated and proposed, but the therapeutic outcome in these patients is low, as a result of drug resistance [2]. The gradual accumulation of genetic and epigenetic alterations, as well as environmental factors, destabilizes the DNA, and leads to an abnormal gene expression stemming from the coordinated deregulation of transcription factors (TFs). In addition, mutations in genes encoding protein kinases are critical for the onset of carcinogenesis.

Thus far, most efforts focus on the detection of differentially expressed genes (DEGs) between cancerous and normal lung tissue. Nevertheless, the reported DEGs vary from study to study, depending on biological differences in the profiled samples (e.g., sex, mutation status, subtype, stage, etc.), as well as differences in sample numbers or the computational methodology followed [3]. One intriguing issue not yet contemplated is the analysis of the genes and signaling pathways being simultaneously deregulated across molecularly heterogenous subtypes in lung cancer, including lung adenocarcinomas and squamous cell carcinomas. Challenging as it may seem, the analysis of pooled, diverse datasets of lung cancer could reveal the commonly deregulated signaling pathways across these tumor entities. Such information could be very useful for the treatment of what really should be viewed as many different diseases.

Extracting large-scale data from the Gene Expression Omnibus (GEO) offers extensive possibilities of simultaneously managing and analyzing multiple gene signatures. Collecting gene expression profiles of co-deregulated genes (co-DEGs) at the transcriptional level can help us understand the complete activation and overlapping of molecularly diverse oncogenic paths. Abnormal patterns of gene expression can also be used for the detection of disease biomarkers, as well as for therapeutic purposes.

There are many genes being differentially expressed across distinct histological or molecular subtypes in lung cancer [4,5,6,7,8,9,10,11]. Nevertheless, the sensitivity and specificity of these biomarkers is not always necessarily adequate, urging the need to identify an updated panel of genes that can be used as better diagnostic and preventive biomarkers, or even as therapeutic targets. In addition, the co-DEGs across distinct lung tumors vary and, to our knowledge, their role within signaling networks or their transcriptional regulatory mechanisms has been poorly investigated. A recent new multivariate method called the Characteristic Direction (CD), can be used to compute signatures using the orientation of the separating hyperplane from a linear classification scheme to define a direction that characterizes differential expression [12].

The purpose of this study was to use the CD method to identify co-DEGs across various independent datasets in lung cancer. We used GEO datasets to annotate and extract gene expression signatures, and validated our results in the TCGA-LUAD and TCGA-LUSC datasets, as well as in the Human Protein Atlas, aiming to better understand the links between co-deregulated genes, drugs and lung cancer. Importantly, we aimed to identify the “hubs” in the gene networks composed of transcription factors and protein kinases differentially expressed in lung cancer. We also intended to identify drugs (or drug combinations) targeting these hubs. Such drugs could thus be used as new and effective treatment regimes. The lists of co-deregulated gene signatures that we propose can provide further insights into lung carcinogenesis.

## 2. Results

Using a Systems Biology approach, we extracted the co-deregulated signatures (i.e., signatures containing the same deregulated genes present in at least two independent studies) from 25 independent GEO datasets and classified them as: (1) lung cancer vs. healthy tissue; lung cancer (tissue or cell lines) with a single gene (2); or single-drug (3) perturbation (Appendix A). We asked whether signature similarity within and across these three categories could recover prior knowledge and discover new connections. To globally assess associations between signatures within each category, we computed the signatures using the CD method, and compared ranked signature associations with prior knowledge. We concluded such lists of co-deregulated genes in each category after stringent filtering and excluding the genes that were deregulated in a single study. These lists contained 20 co-upregulated and 25 co-downregulated genes in LC versus the normal tissue; 333 co-upregulated and 528 co-downregulated genes after single-gene perturbation; and 459 co-upregulated and 439 co-downregulated genes after single-drug perturbation (Appendix A). In each category we identified the TFs, PPIs, and kinases accountable for the observed changes in the mRNA expression of these co-DEGs, the drugs that suppress or induce these co-DEGs, respectively, and finally the biological pathways in which the abovementioned genes are involved.

Next, we sought to identify which of these upstream regulators are most probably responsible for the deregulation in the expression of the identified gene lists in lung cancers. To this end, in each category we identified the phosphorylation reactions possibly being carried out by upstream regulatory kinases. We also investigated the drugs that suppress over-expressed genes, or those that induce the expression of under-expressed ones. The top TFs and protein kinases from each category of co-DEGs were classified based on the highest value of a combined score of the *p*-value and the z-score (Appendix A).

### 2.1. Co-Deregulated Genes in Lung Cancers 

We analyzed two GEO datasets to detect the co-DEGs in lung cancer samples against their adjacent healthy tissue (Appendix A). The co-upregulated genes were mainly enriched in the “immune response-regulating cell surface receptor signaling pathway” (GO biological process) and the “Sec61 translocon complex” (GO cellular component), as well as “platelet-derived growth factor binding” and “cell adhesive protein binding involved in bundle of His cell–Purkinje myocyte communication” (GO molecular function) (Figure 1a and Appendix A).

We further explored differences in the expression of individual genes within the top enriched terms of the co-up- or co-downregulated genes across different immune and molecular subtypes in lung adenocarcinoma (LUAD) and lung squamous cell carcinoma (LUSC) datasets from the Cancer Genome Atlas (TCGA), respectively. Specifically, the immune subtypes we explored were: C1 (wound healing); C2 (IFN-gamma dominant); C3 (inflammatory); C4 (lymphocyte depleted); C5 (immunologically quiet); and C6 (TGF-b dominant), as previously described by Thorsson et al. [13]. The molecular subtypes of LUSC contained basal, classical, primitive, and secretory lung tumors. Among the top co-upregulated genes, we examined “anterior gradient 2, protein disulphide isomerase family member” (AGR2), “alpha-2-glycoprotein 1, zinc binding” (AZGP1), “small cell lung carcinoma cluster 4 antigen” (CD24), “collagen, type I, alpha 2” (COL1A2), “collagen, type III, alpha 1” (COL3A1), “collagen triple-helix repeat containing 1” (CTHRC1), “desmoplakin” (DSP), and “joining chain of multimeric IgA and IgM” (JCHAIN). Similarly, among the top co-downregulated genes, we explored “brain-expressed, X-linked 1” (BEX1); “chemokine (C-C motif) ligand 2” (CCL2); “C-type lectin domain family 3, member B” (CLEC3B); “cysteine-rich intestinal protein 1” (CRIP1); “epithelial membrane protein 2 (EMP2)”; “fatty acid-binding protein 4, adipocyte (FABP4)”; “Fc fragment of IgG, low-affinity IIIb, receptor (CD16b)” (FCGR3B) and “ficolin (collagen/fibrinogen domain containing) 1” (FCN1). Interestingly, we found significant differences in the expression of these genes across the different molecular and immune subtypes in lung cancer. For example, AGR2, CD24, COL3A1, CTHRC1, DSP and JCHAIN were upregulated in basal LUSC tumors. Similarly, AZGP1, COL1A2, COL3A1 and CTHRC1 were upregulated in “TGF-b dominant” (C6) LUSC samples (Figure 2).

The co-downregulated genes, on the other hand, were mainly enriched in “peptidyl-cysteine S-nitrosylation”, “leukocyte aggregation” (GO biological process) (Figure 1b and Appendix A), “cytolytic granule” (GO cellular component), “RAGE receptor binding”, “arachidonic acid binding”, and “icosatetraenoic acid binding” (GO molecular function).

KEGG enrichment analysis highlighted the involvement of the co-upregulated genes in the pathways: “Relaxin signaling pathway” and “ECM–receptor interaction” (Figure 1c and Appendix A). As for the co-downregulated genes, these were overrepresented in “Graft-versus-host disease”, “IL-17 signaling pathway” and “Allograft rejection” (Figure 1d and Appendix A).

We then constructed the PPI networks containing the critical hub proteins in lung carcinogenesis, as they potentially regulate the expression of the co-DEGs. The responsible upstream regulators for the co-upregulated genes involved CTCF, EZH2, TCF3, SOX2, RAD21, FOSL2, IRF1 and SMC3 (TFs) and CDK1/2, GSK3B, PLK1, AKT1, CDK4, DNAPK, ATM, CSNK2A1 and MAPK14 (kinases) (Figure 3a and Appendix A). The co-downregulated genes on the other hand, were significantly associated with the TFs HDAC2, PPARG and RUNX1 and the kinases GSK3B, MAPK14, MAPK3, ERK1/2, CSNK2A1 and CK2ALPHA (Figure 3b and Appendix A).

Apart from selected genes, we also successfully validated the co-deregulated gene signatures in lung cancer (termed “UP genes” and “DOWN genes”, respectively) using the TCGA-LUAD and TCGA-LUSC datasets. Both gene expression signatures confirmed their significant difference in the LUAD and LUSC TCGA datasets, compared to the control tissues. As controls, we used normal lung samples from both the TCGA and GTEx projects (Figure 4a,b).

In addition, we selected specific hub proteins to verify their gene expression, using GEPIA2 [14]. We further compared disease-free survival in patients with high expression in signatures composed of the hub TFs FOSL2, CHD1, SOX2, KLF4, TP63, STAT3 and MYC (Figure 4c) or the kinases MAPK14, GSKB3, CDK4, CDK1, MAPK3, CSNK2A1, HIPK2, and MAPK8 (Figure 4d) against those with low expression and found that, although high expression of the TF hubs’ signatures was not associated with survival, the high expression of the signature composed of the eight hub kinases was significantly associated with disease-free survival (*p* = 0.0044, Log-rank). Of these, GSK3B, CDK1 and CDK4 were positively correlated with LUAD patients’ disease-free survival (DFS), whereas MAPK3, CSNK2A1, HIPK2 and MAPK8 were correlated with LUSC patients’ DFS (Figure 4d).

The TFs MYC and TP63 were among the main hubs, and their upregulated levels were mainly verified in LUSC (Figure 4e,f). MYC stained high in 10% of the cases using IHC data from the Human Protein Atlas and its high expression was an unfavorable marker in lung cancer (*p* = 0.0049, Log-rank) (Figure 4g). On the other hand, TP63 stained high in 40% of the cases (Figure 4f) and its high expression correlated with better patient survival (*p* = 0.052, Log-rank) (Figure 4h). Similar analyses were performed for other hub proteins, including CEBPB, RUNX1, GATA1, etc.

### 2.2. Drug Repurposing against Lung Cancer 

Drug repurposing is a strategy for identifying new uses for approved or investigational drugs that are outside the scope of the original medical indication. Here, using Connectivity Map (CMap) and the identified co-DEGs, we found two existing drugs mainly targeting the co-upregulated genes (valproic acid and betonicine) and one drug (astemizole) against the co-downregulated genes (Figure 1e,f and Appendix A) [15,16,17,18,19].

### 2.3. Signatures of Co-Deregulated Genes in Single-Gene Perturbation Experiments

We extracted the co-DEG signatures from 11 single-gene perturbation studies that were enriched in biological processes involving “positive regulation of Tau protein kinase activity”, “regulation of Tau protein kinase activity”, “negative regulation of plasminogen activation”, “cytoplasmic translation” and “co-translational protein targeting to membrane” (Figure 5a and Appendix A). A remarkable finding is the modulated expression of Tau protein, which normally penetrates to the brain tissue, taking part in the organization of the axial microtubules of nerve cells. This correlated relationship is admittedly anticipated, as similar risk factors such as cell ageing, irregular cell cycle and cell death due to DNA damage turn out to be present in both cancer development and neurological disorders. It has recently been proposed that Tau protein appears in smooth muscle cells of the vascular and airway systems of the lung [15], and there is a positive correlation between Tau protein and the evolution of pulmonary adenocarcinoma [16]. The common cellular pathogenetic events between these two clinically distinct diseases require further study in light of molecular cross-linking given the tremendously high rate of metastatic tendency of lung tumors to the brain [17,18]. Subsequently, these genes co-overexpress in the negative regulation of plasminogen activation, a system that has dissimilar gene expression patterns among lung cancer subtypes [19]. They are also enriched in cellular components such as “cytosolic small ribosomal subunit”, “alveolar lamellar body”, “small ribosomal subunit”, and “multivesicular body lumen”, as well as in operating functions such as “aminophospholipid flippase activity”, “aspartic-type endopeptidase inhibitor activity” and “cytochrome-c oxidase activity”. Aminophospholipid flippases are phospholipid transporters responsible for the formation and maintenance of the cytoplasmic asymmetry of membranes, a unique architecture that contributes to various signaling processes such as apoptosis mediated by phagocytes. A high flippase activity has been found to regulate the low surface exposure of phosphatidylserine residues in human cancer cells [20], an altered pattern of which is responsible for immunological imbalance through silencing key inflammatory signals. In this way, the deregulated action of flippases brings about an immune-escape phenomenon of malignant cells, as well as induces immunosuppressive phosphatidylserines [21]. Aspartic-type endopeptidase inhibitor activity assists in the hydrolysis of a peptide bond in the side chains of aspartic acid so that a nucleophilic character is acquired. Dysfunctionality arises from an endogenous abnormal feature of carcinogenesis to disorientate transcription factors from functioning properly.

Our KEGG analysis revealed the enrichment of both classes of co-DEGs (i.e., co-up- and co-downregulated genes), mainly in the “Ribosome” and “Coronavirus disease” pathways (Figure 5c,d and Appendix A). Ribosomal malfunction reflects the central role that ribosomes play in the control of gene expression. Increased ribosomal biogenesis leads to high levels of protein synthesis as a principal point in maintaining and advancing cancer, without the complete mechanism being known yet.

Interestingly, the second pathway hints towards the molecular association of COVID-19 disease with lung cancer [22]. More specifically, these two separate conditions share a common pathogenetic ground, partially aligned with the lungs being vulnerable to COVID-19. Currently, there is evidence under consideration for developing lung malignancy following SARS-CoV-2 infection.

As for the co-downregulated genes, we detected enrichment in “SRP-dependent cotranslational protein targeting to membrane”, “cytoplasmic translation”, “co-translational protein targeting to membrane”, “protein targeting to ER” and “nuclear-transcribed mRNA catabolic process, nonsense-mediated decay” (GO biological process); “cytosolic small ribosomal subunit”, “cytosolic large ribosomal subunit”, “small ribosomal subunit” and “large ribosomal subunit” (GO cellular component); and “ubiquitin ligase inhibitor activity”, “ubiquitin–protein transferase inhibitor activity” and “insulin-like growth factor II binding” (GO molecular function) (Figure 5b). Most successive stages of protein synthesis are disrupted by carcinogenetic processes that create a viable tumor microenvironment by controlling the protein interactions at the cellular level. In addition, the deviation from normal protein functionality is reported to be decisive in carcinogenetic events, as there is a dual pathological targeting of translational and post-translational modification processes, as the above-mentioned results reference. Ubiquitination is the physiological process of protein degradation being involved in signal transduction, and results in morbid effects of the solidification of lung tumors. Furthermore, cancer growth signaling moderated by the IGFBP2/IGF pathway has been extensively studied due to its mitogenic nature. Its complex involvement in a plethora of biological procedures results in an uneven motif of overexpression among patients with a diverse clinical picture. In cancer cells, IGFBP2 expression is topologically imported into the cytoplasm and the nucleus is absent of a nuclear localization sequence. Conflicting points of view do not fully explain the oncogenic or tumor suppressive effect of IGFBP2, stating that more in vitro functional studies are needed.

The TFs RELA and MYC, along with the kinases CDK1 and MAPK14, were significant regulatory mediators of both the co-up- and co-downregulated genes following single-gene perturbation. In addition, PML, MYC, RELA, KAT2A, CEBPB, KLF4, TCF3, SOX2, NELFE and ZMIZ1 (TFs), as well as MAPK8/14, CDK1/4 and AKT1 (kinases), were central hub proteins in the co-upregulatory network. Similarly, MYC, RELA, KAT2A, TAF1/7, CEBPD, KLF4, SOX2, NELFE, and ZMIZ1 (TFs), and MAPK1/3/8/14, ERK2, CDK1/4, HIPK2, and JNK1 (kinases), were central hubs in the co-downregulatory network, respectively (Figure 6a,b and Appendix A).

### 2.4. Signatures of Co-Deregulated Genes in Single-Drug Perturbation Experiments

We reanalyzed 12 independent GEO datasets to identify the co-DEGs in samples undergoing a single-drug perturbation. The co-upregulated genes were enriched in “SRP-dependent co-translational protein targeting to membrane”, “co-translational protein targeting to membrane”, “cytoplasmic translation”, “protein targeting to ER” and “nuclear-transcribed mRNA catabolic process, nonsense-mediated decay” (GO biological process), “cytosolic small ribosomal subunit”, “cytosolic large ribosomal subunit”, “large ribosomal subunit”, “small ribosomal subunit” and “ribosome” (GO cellular component), as well as in “retinal dehydrogenase activity”, “large ribosomal subunit rRNA binding”, “oxidoreductase activity, acting on the aldehyde or oxo group of donors, NAD or NADP as acceptor”, “ubiquitin ligase inhibitor activity” and “ubiquinol-cytochrome-c reductase activity” (GO molecular function) (Figure 7a and Appendix A). Alternated patterns of protein synthesis are observed in controlling their production machinery, as well as the roles the proteins perform. Regarding the molecular processes, the co-upregulated genes were related to biosynthetic biochemical events, and took part in the transmission of intracellular signals. Indeed, in certain cases of non-small cell carcinoma, the expression of retinal dehydrogenase is particularly high and plays a role in the synthesis of increased ATP being used by cancer lung cells [23]. Ubiquinol-cytochrome-c protein reductase is actively implicated in the electron transport chain, but even though it has been suggested as a possible diagnostic biomarker in pulmonary adenocarcinomas [24], its usefulness remains under investigation.

In addition, the co-downregulated genes were enriched in “cytoplasmic translation”, “SRP-dependent co-translational protein targeting to membrane”, “protein targeting to ER”, “nuclear-transcribed mRNA catabolic process, nonsense-mediated decay” (GO biological process); “cytosolic small ribosomal subunit”, “small ribosomal subunit”, “Cytosolic large ribosomal subunit”, “ribosome” and “large ribosomal subunit” (GO Cellular component); and in “RNA binding”, “ubiquitin ligase inhibitor activity”, “C3HC4-type RING finger domain binding”, “ubiquitin–protein transferase inhibitor activity” and “mRNA 5’-UTR binding” (GO molecular function) (Figure 7b). The “C3HC4-type zinc finger” region of the RING protein complex is part of several proteins involved in a variety of cell growth and differentiation functions, justifying its transcriptional regulatory application in lung oncogenesis. This factor has been reported to be important in facilitating cell proliferation in lung tissue, diverting ubiquitination pathways in combination with metastatic penetration [25]. The 5’-UTR mRNA binding process reduces the expression of the five untranslated regions of mature mRNA which are responsible for the gene expression post-transcriptional activities necessary for cellular homeostasis. The non-expression of these regulatory elements provokes the consequent deregulation of the normal expression pattern, enhancing pathogenetic effects in lung cancer.

Furthermore, similar to the single-gene perturbation experiments, KEGG analysis for both co-DEGs in this category, revealed enrichment in “Ribosome” and “Coronavirus disease” (Figure 7c,d and Appendix A).

The TFs MYC, KAT2A and CEBPB seem to be the most significant hubs among the co-upregulated genes, while MYC, PML, ATF2 and E2F1 seem to be the most significant hubs among the co-downregulated genes in single-drug perturbation experiments (Figure 8a and Appendix A). Additionally, MAPK14, CDK1, ERK1/2, HIPK2 and CDK4 were the main hub kinases regulating the expression of the co-upregulated genes, whereas MAPK14, CDC2, DNAPK and CDK4 were the main hubs across the co-downregulated genes in this group (Figure 8b and Appendix A).

## 3. Discussion 

In the present work, we used a Systems Biology approach to detect the co-deregulated genes across different lung cancer studies, orchestrated by an extended network of TFs and kinases. We investigated these co-deregulated gene signatures in-depth and classified them into three different categories, focusing on their upstream regulators, networks, hub proteins and interactions. The new signatures of co-DEGs recovered prior knowledge, but also discovered new connections, adding significant information to the pathobiological basis of lung carcinogenesis.

Assembling the co-DEG signatures in lung cancer and exploring their pathways, we found that they are primarily enriched in immunological destabilization, a major hit in the lung. Immune deregulation is a determining point in the development of tumorigenesis in the lungs, alongside presenting a deteriorating clinical status among patients. Cancerous cells attempt to reverse this protective function by controlling primary cellular signals to deviate from the well-orchestrated immune system. In particular, the perception of the extracellular and intercellular microenvironment is of vital significance for a coordinated immunological response. A similar bioinformatics analysis highlighted the enrichment of the down-regulated genes in angiogenesis in calcium ion binding and cell adhesion [26], whereas the up-regulated genes were significantly enriched in the extracellular matrix disassembly, collagen catabolic process, chemokine-mediated signaling pathway, and endopeptidase inhibitor activity. Our study demonstrates the existence of a strong correlation between the disruption of canonical immunological procedures in co-upregulated and down-regulated genes, respectively. Similar to our results, Yu et al. found enriched critical factors in the regulation of immune responses, inducing tumor growth and metastasis [26]. In a broader context, both studies address the importance of deregulated inflammatory responses in lung cancer, and any differences should be attributed to the different sample numbers analyzed in each GEO dataset or the methodological plan followed.

The co-upregulated genes negatively affect cancerous transformation, as their presence at key topological sites of protein synthesis and transcription has a substantial impact on the expression of genetic information. The enriched terms referring to the molecular function also showed a strong association with lung cancer. In addition, platelet-derived growth factor (PDGF) signaling contributes to a wide variety of developmental procedures, and multiple abnormalities have been documented to occur during lung tumorigenesis. Interestingly, “Relaxin signaling pathway” and “ECM–receptor interaction” were found to be enriched in KEGG pathway analysis. Relaxin is a peptide hormone that acts both in an autocrine and paracrine manner by stimulating the nitric oxide (NO) guanosine pathway after binding to its RXFP1 receptor [27]. Relaxin is involved in the processes of lung structure and remodeling [28], but it seems that its upregulation is also associated with tumor invasion [29]. Furthermore, enhanced RXFP1 activation mediates anti-apoptotic and angiogenetic events [30] through ECM degradation [31], providing several implications in the development of chemoresistance [29]. In addition, the metastatic tendency of lung cancer is of high concern owing to the cellular structural changes taking place within the ECM architecture, as the process of malignant establishment continues to occur.

The expression of co-downregulated genes is directly related to dysregulated immunological pathways, ensuring that immune tolerance to malignant cells is upheld and no antioncogenic action occurs. At a biochemical level, the addition of NO to cysteine residues (S-nitrosylation) has been demonstrated to amplify tumor progression and provide no effective response to anticancer medication [32]. Tumor-infiltrating leukocytes are known to have a major, yet antagonistic, role in cancer surveillance and appear to disrupt epithelial tissues during inflammation. More specifically, angiogenetic events happen during the onset of tumorigenesis, as the infiltrated leukocytes produce pro-angiogenic agents that maintain a favorable tumor microenvironment. Likewise, cell adhesion molecules are the main membrane glycoproteins that regulate immune cell migration and activation, through which malignant cells can spread to secondary tissues. In addition, it was recently proposed that leukocyte aggregates are associated with high thrombosis risk in lung cancer patients [33], corresponding to the fact that leukocytes accumulate in the tumor microenvironment in order to facilitate their progression and survival. In this context, it is assumed that a high leukocyte aggregation triggers a metastatic profile of lung cancer to distant sites, and is correlated with fatal cardiovascular incidents. Indeed, in a recent observational study, cancer patients were shown to have a high mortality risk due to cardiovascular disease [34]. Thus, the co-downregulated genes were mainly enriched in “cytolytic granule” (GO cellular component) and took part in molecular processes such as “RAGE receptor binding”, “arachidonic acid binding”, “icosatetraenoic acid binding”, “eicosanoid binding” and “Toll-like receptor binding”.

The co-downregulated genes were enriched in “RAGE receptor functions of glycosylation end products”, which are mainly presented in diabetic pathogenesis, but the RAGE receptor is associated with lung carcinogenesis through the induction of multiple intracellular oxidative stress pathways [35]. Nevertheless, RAGE receptors appear to exhibit multiligand characteristics by activating responses in inflammatory conditions, and they are involved in the maturation process of dendritic cells, T cell proliferation, and polarization to CD4+ cells. At the same time, there is a functional relationship with the TLR receptor family through the initiation of innate immune responses because of the joint interaction ligands. TLRs have regulatory features to mediate cancer cell growth, secondary tissue penetration, angiogenesis, and cancerous conversion in different molecular mechanisms.

In addition, the metabolic pathway of arachidonic acid is linked to the biological basis of lung cancer due to the involvement of biosynthetic cyclooxygenases (COX-2) and phospholipase A2 (PLA2) in the MAPK/ERK and EGFR pathways [36]. Regarding the binding function of eicosapentaenoic acid, omega-3 fatty acid supplements are well-known to have anti-inflammatory properties against cancer cachexia syndrome [37], playing a beneficial role in the inhibition of cell proliferation and the reversal of the arachidonic acid metabolism [38]. This correlates with previous findings, reinforcing the need for the co-prescription of supplements in lung cancer patients. Finally, similar effects are revealed in eicosanoid acid-binding processes that act as lipid mediators of arachidonic acid, the production of which is higher both in tumor cells and their surrounding microenvironment [39,40], hinting towards a new treatment approach that targets the biochemical cycle of lung cancer cells.

Diabetic cardiomyopathy may be associated with lung carcinogenesis via the activation of the AGE–RAGE signaling pathway. The disease also associates with the intracellular stimulation of oxidative stress, whereas the RAGE receptor modulates immune responses in the lungs [41]. Pathogenic Salmonella infection is not correlated with tumorigenesis in the lung, but the organ remains vulnerable to infections. This finding warrants further investigation in the lung microbiome to better understand the properties of its drug resistance.

ECM is a highly complex structure of biologically active macromolecules that control key cellular functions. The interaction between ECM and tumor cells underlines the tendency of the latter to infiltrate the tumor microenvironment. The dynamic and organizational structure of ECM is considered to be the basis of cellular behavior. Tumor-associated macrophages (TAMs) and neutrophils also play a key role in the onset of carcinogenesis, performing phagocytic processes (“phagosome”) through lysosomal digestion. In the majority of different cancer types, antiphagocytic signals are expressed to escape immune surveillance, and thus malignant proliferation is not eliminated.

The terms that we found to correspond to the KEGG pathway analysis for the co-downregulated genes refer to certain immunological procedures taking place during histocompatibility and the development of an acute inflammation response. After all, lung is a target organ of the graft-versus-host disease response due to the consecutive emergence of cytotoxic T-lymphocytes and NK cells influenced by the cytokines IL-1/TNFα [42]. The interleukin 17 (IL-17) signaling pathway may be enriched because of its participation in tumor-associated inflammation. In fact, IL-17 is a pro-inflammatory cytokine [43] that contributes to the metastasis of lung cancer cells [44].

PPI networks contribute to a further understanding of the transcriptional machinery and phosphorylation reactions driving the pathobiological events of lung carcinogenesis. Here, we sought to investigate the interactions among the co-DEGs in lung cancer and certain hub proteins as critical mediators in the signaling pathways. We highlight EZH2, CTCF, RAD21, HDAC2, RUNX1 and PPARG as the principal hubs closely connected to this disease. Of these, we emphasize on the enhancer of Zeste Homologue 2 (EZH2), the CCCTC-binding factor (CTCF), and the RAD21 Cohesin Complex Component (RAD21), across the main co-upregulated TFs.

EZH2 possesses a crucial role in chromosomal remodeling, as well as in the regulation of the silencing of several tumor suppressor genes and those involved in immune cell development [45]. The overexpression of EZH2 allows cancer cells to divide uncontrollably and plays an important role in the acquired chemoresistance of cancer cells [46].

CTCF has multiple functional roles in chromosomal interactions, leading to gene expression, while its rs60507107 variant correlates with an increased risk of lung cancer [47]. As a chromatin architecture mediator, it epigenetically regulates transcription and CTCF-binding alterations can be considered as epigenomic signatures of cancer development. A cancerous transition mechanism in lung fibroblasts relies upon the deregulated expression of Rb2/p130, which is controlled by CTCF, and has been shown to promote the progression and recurrence of lung cancer after treatment. Thus, CTCF upregulation promotes carcinogenetic effects, as it ultimately organizes the genome structure and can alter gene expression.

RAD21 normally participates in sister chromatid cohesion and separation when needed during transcription, DNA replication, or DNA damage repair mechanisms. RAD21 knock down shows resistance to DNA-damaging chemotherapeutic drugs in vitro [48], and its increased expression is evident in poorly differentiated lung cancers due to its contribution to the regulation of the cell cycle [49].

On the other hand, among the co-downregulated transcription factors in lung cancer, we focused on RUNX1, histone deacetylase 2 (HDAC2), and peroxisome proliferator-activated receptor gamma (PPARG).

RUNX1 participates in different hallmarks of cancer, such as the developmental differentiation of multiple human cell lines. Its downregulation was recently linked with poor lung cancer patient survival [50], while its altered methylation pattern is used as a biomarker in non-small cell lung cancer (NSCLC) [51]. Furthermore, RUNX1 is known to inhibit the transcription of YAP [52] in breast cancer, a molecular interplay that leads to immunosuppressive events during lung tumorigenesis [53].

HDAC2 functions as a part of large multiprotein complexes that repress transcription through the deacetylation of lysine residues on the N-terminal part of histones. In NSCLC, HDAC2 knock down correlates with a low expression of fibronectin (FN) [54] and enhances the metastatic potential of lung cancer cells [55]. The functional alteration and imbalance in epigenetic modulation affects major pleiotropic cellular events and growing interest has been raised towards clinical utility.

A decreased expression of PPARG is a prognostic marker for NSCLC. PPARs regulate cancer-relevant processes, such as cell differentiation, proliferation and apoptosis [56,57]. In colorectal cancer, emerging evidence has shown that PPARG signaling is downregulated due to the regulatory actions of EZH2 and HDAC1 [58]. In fact, our results come in accordance with this statement and we, therefore, consider that the axis between PPARs, EZH2 and HDACs is a novel transcriptional interplay underlying lung carcinogenesis, and thus needs further investigation.

Moreover, we detected specific repurposing drugs to potentially extend the therapeutic opportunities in lung cancer. A beneficial effect in lung cancer treatment could result from the pharmacological regulation in the expression of co-DEGs. Here, we paid specific attention to valproic acid (VPA), betonicine, and astemizole.

Valproic acid is a histone deacetylase (HDAC) inhibitor used in the treatment of epilepsy because of its mood-stabilization properties. As such, VPA can regulate the expression of genes in small cell lung cancer (SCLC), as well as balance apoptotic modulators and suppress cell growth via the activation of the Notch 1 signaling pathway [59]. VPA was also recently reported to effectively inhibit the proliferation of lung cancer cells in vivo, when combined with arsenic trioxide, via the induction of apoptotic signals [60]. In addition, VPA was shown to improve a second-line regimen of small cell lung carcinoma in preclinical models, opening new prospects for improved therapies [61]. The anticancer activity of VPA was also shown in non-small cell lung cancer (NSCLC) cell lines when combined with a new cyclin-dependent kinase (CDK) inhibitor (P276-00) [62]. Therefore, our findings corroborate previous reports that VPA is a promising new molecularly targeted therapeutic approach for the treatment of lung cancer.

Betonicine is a pyrolidine alkaloid isolated from Achillea millefolium that inhibits bacterial signaling molecules being used as an adjuvant to anti-infective treatments. It targets several enzymes involved in drug metabolism (mainly CYPs). Some types of betonicine’s inhibitory activity are against a and β glucosidases, as well as β-manosidase and eukariotic DNA polymerases. The toxic effect of pyrolidine alkaloids is generally manifested in the liver [63,64]; nevertheless, betonicine’s anti-tumor effects in lung cancer are widely unknown.

As for the repurposing drugs that seem to induce the co-downregulated genes in lung cancer, here we highlight astemizole. This drug has been commonly used in the treatment of allergies as an antihistamine, but it was reported to cause arrythmias when administrated in high doses. Astemizole targets several proteins involved in tumor cell proliferation [65], and its combined pharmacological action with gefitinib was recently shown to have promising results in lung cancer [66]. Our findings corroborate these reports and strongly suggest that astemizole could be used as a repurposing therapeutic drug for lung cancer patients. Together, these findings identify the necessity of integrating repurposed drugs into therapeutic efforts to treat lung cancer.

All in all, here, we analyzed pooled data sets across heterogenous subtypes of lung cancer and identified the top co-DEGs and their hubs along with targeted therapeutic drugs. Our results show that all of the three signature categories recover prior knowledge associations between genes, drugs, and diseases.

## 4. Materials and Methods

### 4.1. GEO Data Extraction and Filtering 

We first assorted 24 studies from the Gene Expression Omnibus (GEO) to extract differentially expressed genes using GEO2Enrichr [67,68]. The studies were classified into three categories: (1) those assessing gene expression profiles in LC vs. healthy samples (2 GEO studies), (2) those subjected to single-gene perturbation (10 GEO studies), and (3) those observing any therapeutic effect through single-drug perturbation (12 GEO studies) in lung cancer. The selection process was strictly focused on gene expression studies containing tissue samples or cell lines of human or mouse origin. The standard naming of genes, diseases, and drugs was provided as an autocomplete option in the submission forms created from HGNC [68], Disease Ontology [69], and DrugBank [70], respectively.

The corresponding mined data sets were as follows: GDS4794, consisting of 23 lung carcinoma tissue samples and 2 normal tissue samples; GDS3837, containing 60 samples of healthy tissue and 60 samples of non-small cell lung cancer from women; GDS5418, containing 4 normal A456 cell lines and 4 SRC-/- A456 cell lines; GDS5391, consisting of 4 NCI-H1299 and NCI cell lines samples (2 for each line) of pulmonary adenocarcinoma not expressing the protein kinase of tyrosine PTK7 and 4 normal samples (control); GDS2489, consisting of 18 normal samples of lung airway epithelial cells and 26 samples of the same cell line from smokers; GDS3510, composed of CL1-5-derived cells overexpressing Claudin-1 (CLDN1) derived from pulmonary adenocarcinoma tissue; GDS3029, having 27 small cell lung cancer cell samples resistant to Bcl-2 antagonist ABT-737 and 7 normal samples (control); GDS3826, containing 5 transgenic SP-C/c-raf lung samples and 10 healthy samples; GDS5206, consisting of 9 samples of A549 epithelial cell line from pulmonary adenocarcinoma and 9 healthy controls; GDS4847, containing 40 lung samples from B6C3F1 mice that were given chloprene and 10 normal tissue samples; GDS5067, consisting of 5 control samples and 9 A549 cell line samples treated with oligopiperazines (OOPs) BB2-125, BB2-162 and BB2-282; GDS3101, composed of A459-derived cis-platinum-resistant cells, 3 samples that were treated with cis-platinum, and 3 more samples that were not; GDS5496, consisting of 2 NCI-H441 lung-derived cancer cells expressing hsa-miR-365-2 and 2 normal samples; GDS3825, analyzing 5 transgenic SP-C/c-raf cancerous lung cells, 5 non-transgenic SP-C/c-raf samples and 5 completely normal samples; GDS5201, containing 4 tumor samples resulting from activation of the Wnt/beta-catenin signaling pathway in combination with the expression of KRAS protein and 2 samples of normal lung tissues; GDS2966, consisting of A549 lung cancer cells, of which 2 samples were exposed to resveratrol and 2 more remained intact; GDS5648, consisting of 5 K-RAS-driven mouse tissue and 5 controls; GDS1649, containing 15 samples of mouse lung cancer and 29 samples that were subjected to iodine urethane; GDS3321, consisting of 4 mouse lung tumors which had undergone transgenic intervention in alveolar epithelial cells to overexpress c-Myc proto-oncogene and 4 control samples; GDS2958, using 2 samples of HCC827-derived cells with inactive PTEN inhibitor and 2 normal samples; GDS2298, containing 7 samples of NSCLC-derived cell lines sensitive to Gefitinib and 11 control samples; GDS4840, containing 3 samples of CCR5-/-mutant mouse-derived lung tissue and 3 samples from normal tissues; GDS2604, consisting of 14 samples from epithelial and mesothelial lung cancer cell lines and 13 control samples; GDS5247, using 3 samples of H460 parent-derived lung cell line and 3 samples of the same cell line with cis-platin resistance; GDS2499, containing 3 cell samples were cultured in mannitol, 3 samples cultured in actinomycin D, and 6 maintained in saffron PCI-2050 (Appendix A).

We re-processed the extracted expression signatures to filter their quality and check data integrity, as previously explained in detail [3,66]. We also quantified batch effects with variance component analysis [71] and corrected them with surrogate variable analysis (SVA) [68].

### 4.2. Differentially Expressed Genes and Co-DEGs

Genes presenting a differential expression profile (cutoff of 500 genes) were estimated based on the Characteristic Direction (CD) algorithm [12]. The GEO2Enrichr extracted gene sets of the over- and under-expressed genes were manually sorted based on the CD metric. In the event of single-gene perturbation (i.e., knock out, knock down, knock in, RNAi, or overexpression) experiments, the differential expression signatures were considered against their normal (wt) alleles. Regarding single-drug experiments, DEGs were calculated against the non-treated cells or tissues. Differentially expressed genes that were found only in one study were excluded from further analysis. For each group, the DEGs that were identified between at least two independent studies were termed as “co-upregulated” or “co-downregulated”, respectively (co-DEGs).

### 4.3. Upstream Regulators of co-DEGs and Protein–Protein Interaction Networks

Expression2Kinases (X2K) was utilized to investigate the upstream regulatory networks from the co-DEGs signatures within each group. Transcription factors (TFs), intermediate proteins and protein kinases participating in the regulation of transcriptional and expression processes of the inputted co-DEGs were produced as previously described [72]. X2K detects the expected elements within each category using an integrated promoter analysis of ChIP-X to construct a complete network between TFs and known protein interactions [73]. Kinase enrichment analysis was used to locate the upstream protein kinases, which are critical for carrying out phosphorylation reactions that regulate the expression of the identified co-DEGs. We then used Enrichr [74] to create protein–protein interaction networks (PPI) for each group of co-DEGs, involving the implicated TFs, kinases and their intermediated proteins, indicating the nodes (genes) and edges (lines) in each network. The interacting proteins along with the phosphorylation paths involved were also included for the visualization purposes of each PPI network.

### 4.4. GO and KEGG Enrichment Analysis

We performed Gene Ontology (GO) enrichment analysis to reveal the overrepresented GO terms, focusing on the molecular-level activities performed by the top co-DEGs, the locations relative to the cellular structures in which their gene products perform their functions, and the biological processes accomplished by multiple molecular activities using Enrichr [74].

In addition, we used the Kyoto Encyclopedia of Genes and Genomes (KEGG) [75,76,77] to assess the pathway information of the top co-DEGs in each group. Results were sorted using a combined score, defined as c = log(*p*)*z, i.e., the log of the BH-adjusted *p*-value from the Fischer’s exact test, multiplied by the z-score of the deviation from the expected rank. An adjusted *p*-value (adj-*p*) of 0.05 was used as the threshold of statistical significance.

### 4.5. Connectivity Map Analysis 

Connectivity Map (CMap, https://clue.io/cmap, accessed on 21 September 2021) is a large-scale database including shareable data on pharmaceutical agents [78] and substances with a previously defined pharmacological action. These are statistically correlated with provided gene expression signatures, and could potentially induce or reverse LC based on this. We used CMap to identify repurposing drugs for lung cancer based on a connectivity score ranging from -1 to +1. The given range indicates that values closer to +1 have a positive connectivity, as drugs enhance the carcinogenic effects in the lung. In contrast, values closer to -1 have a reversable gene–drug relationship, suppressing the growth of lung cancer cells. We submitted the lists of co-DEGs (over- and under-expressed genes related to lung cancer), and found a group of substances that act reversibly on the studied expression patterns of the co-DEGs. A hypergeometric probability test was used to associate drugs with disease.

### 4.6. Validation of the Co-Deregulated Gene Signatures and Hub Genes in the TCGA and the Human Protein Atlas

We validated the calculated expression signatures in lung cancer after extracting the read counts of RNA-seq data from the Cancer Genome Atlas of lung squamous cell carcinoma (TCGA-LUSC) (486 tumor samples and 338 controls) and the lung adenocarcinoma (LUAD) (483 tumors and 347 controls) datasets using the Genomic Data Commons data portal (https://portal.gdc.cancer.gov/, accessed on 23 January 2022). We then normalized the read counts to log_2_ (TPM + 1) values, as previously described [79]. To increase the sample number of the controls, the TCGA normal data were matched with normal lung samples from the Genotype-Tissue Expression (GTEx) project (https://gtexportal.org/home/, accessed on 23 January 2022). The expression levels of the “UP genes” signature, composed of 54 genes, and of the “DOWN genes” signature, composed of 48 genes, were explored with Limma [80] using log_2_FC = 1 and q-value = 0.01 as thresholds of significance.

In addition, we explored the expression of the top deregulated genes in lung cancer across different molecular subtypes in LUSC (basal, classical, primitive, secretory), as well as different immune subtypes in LUSC and LUAD (C1, wound healing; C2, IFN-gamma dominant; C3, inflammatory; C4, lymphocyte depleted; C5, immunologically quiet; C6, TGF-b dominant) [13].

To further investigate the hub genes and associate their expression with patient survival, we used the Gene Expression Profiling Interactive Analysis 2 (GEPIA2) [14] and immunohistochemistry (IHC) data from the Human Protein Atlas (HPA) [81]. Regarding the IHC protocol that was followed, in brief, FFPE sections (4 μm) were heated at 50 °C overnight. Then, they were deparaffinized in xylene and rehydrated in graded ethanol to distilled water. During hydration, a 5 min blocking for endogenous peroxidase was completed in 0.3% H_2_O_2_ in 95% ethanol. Prior to immunostaining, the sections were immersed in 10mM citrate buffer (pH 6.0), rinsed in Tris-buffered saline (TBS) and subjected to heat-induced epitope retrieval (HIER) using a pressure boiler. Sections were then incubated overnight at 4 °C with mouse monoclonal antibodies (mAbs) against MYC (1:20, Atlas Antibodies Cat#HPA055893, RRID: AB 2682960), TP63 (1:75, Atlas Antibodies Cat#HPA006288, RRID: AB_1080334). The UltraVision LP HRP polymer^®^, Ultra V Block and DAB quanto substrate system^®^ (Thermo scientific, La Jolla, CA, USA) were used for detection. Finally, slides were rinsed in tap water, counterstained with hematoxylin, dehydrated in grade ethanol and cover-slipped. Slides were then assessed independently by two observers.

### 4.7. Patient Survival Analysis

Disease-free survival analysis and corresponding maps for the patients in the TCGA-LUSC and TCGA-LUAD datasets were further constructed using two gene signatures: one containing the hub TFs (MYC, STAT3, TP63, KLF4, SOX2, CHD1 and FOSL2) and another containing the hub kinases (JNK1, HIPK2, CSNK2A1, MAPK3, CDK1, CDK4, GSK3B, ERK1 and MAPK14). For the Kaplan–Meier curves we used hazard ratio (HR), which was calculated based on the Cox PH model and a 95% confidence interval (CI). For the survival maps, we used an FDR-adjusted *p*-value = 0.05 as the threshold of significance and median value cutoff. All experiments were performed in accordance with the TCGA relevant guidelines and regulations.

## 5. Conclusions

Overall, the study of commonly deregulated pathways and molecular interactions reveals the dynamic processes taking place during lung carcinogenesis.

## Figures and Tables

**Figure 1 ijms-23-10933-f001:**
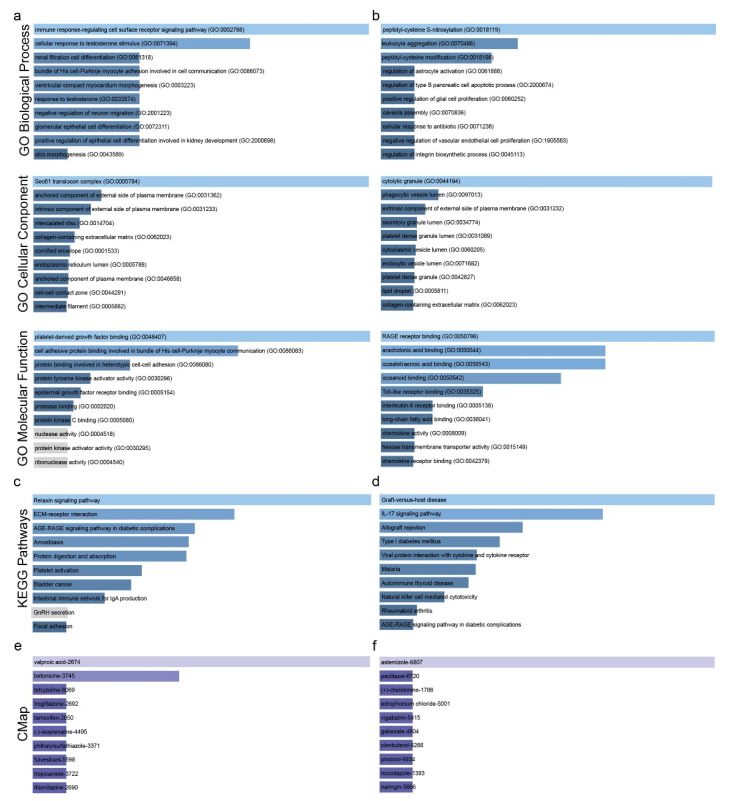
GO enrichment of the co-upregulated (**a**) and the co-downregulated (**b**) genes in lung cancer vs. the normal tissue. KEGG enrichment for the co-upregulated (**c**) and the co-downregulated (**d**) genes in lung cancer vs. the normal tissue. Repurposing drugs (CMap) targeting the co-up- (**e**) and co-downregulated (**f**) genes in lung cancer. Bar graphs were sorted using a combined score between the Benjamini–Hochberg (BH)-adjusted *p*-value and the z-score of the deviation from the expected rank. Each bar’s length shows the significance of the corresponding term, being relative to the color brightness of each bar.

**Figure 2 ijms-23-10933-f002:**
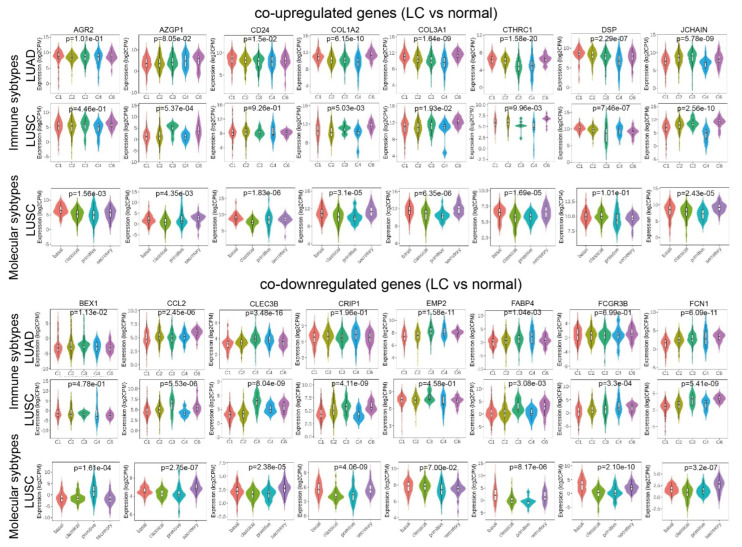
Expression of the top co-up- or co-downregulated genes across different immune and molecular subtypes of lung adenocarcinoma (LUAD) or lung squamous cell carcinoma (LUSC). The immune subtypes were: C1 (wound healing); C2 (IFN-gamma dominant); C3 (inflammatory); C4 (lymphocyte depleted); C5 (immunologically quiet) and C6 (TGF-b dominant), as defined in Thorsson et al. [13]. The molecular subtypes of LUSC contained basal, classical, primitive, secretory tumors. LUAD immune subtypes, n = C1,83;C2,147;C3,179;C4,20;C6,28. LUSC immune subtypes, n=C1,275;C2,182;C3,8;C4,7;C6,14. LUSC molecular subtypes, n=basal 42, classical 63, primitive 26, secretory 39.

**Figure 3 ijms-23-10933-f003:**
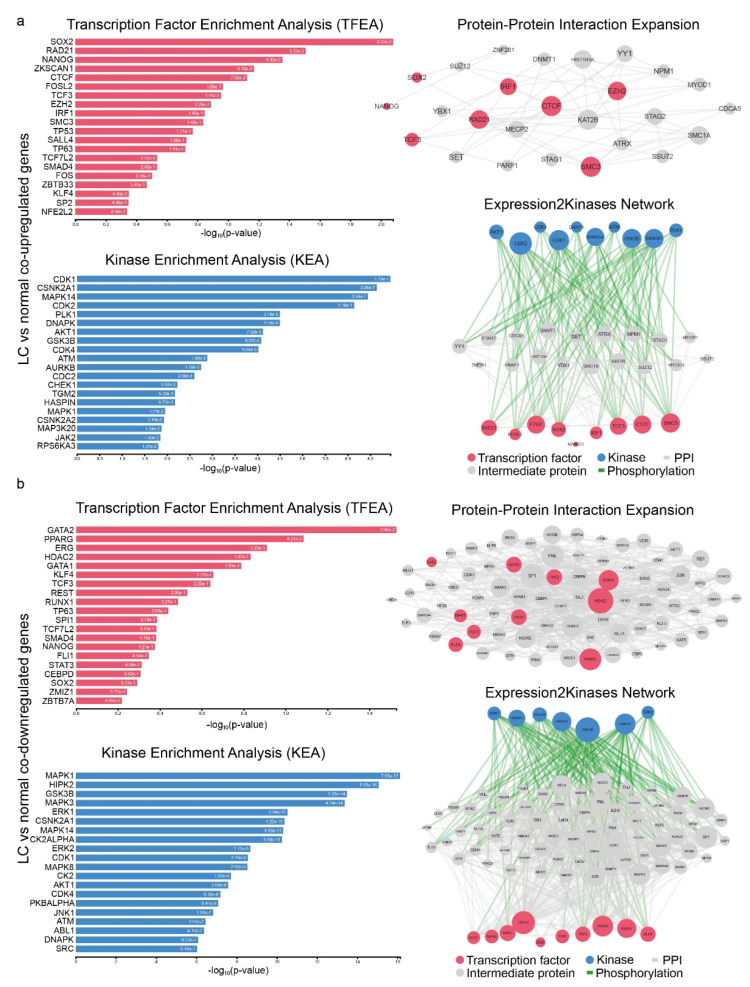
Upstream regulatory networks for co-upregulated (**a**) and co-downregulated (**b**) gene signatures in lung cancer vs. the normal tissue. The networks depict transcription factors (TFs, red nodes), intermediate proteins (gray nodes), and kinases (blue nodes). Gray edges indicate PPI interactions and green edges depict kinase-driven phosphorylation events. Node size is relative to expression. Upstream regulatory networks were constructed using X2K.

**Figure 4 ijms-23-10933-f004:**
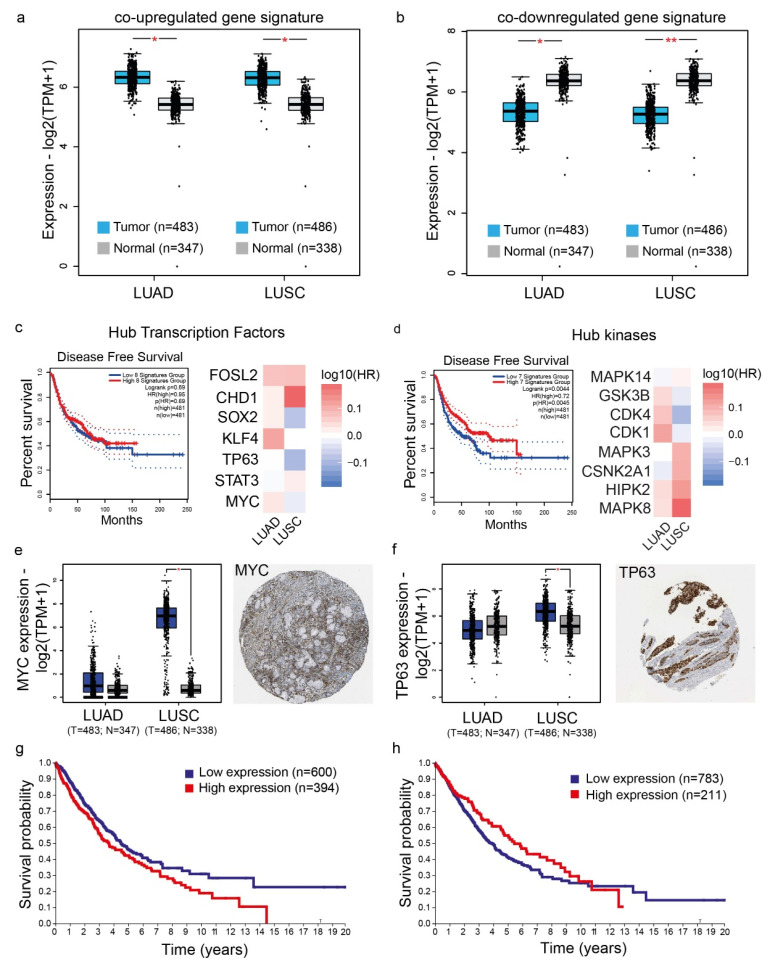
The expression patterns of the co-upregulated (**a**) and co-downregulated (**b**) gene signatures were verified in the TCGA-LUAD and TCGA-LUSC datasets, respectively (LUAD, 483 lung adenocarcinomas (T) and 347 normal (N) samples; LUSC, 486 lung squamous cell carcinomas (T) and 338 normal (N) samples). The significantly elevated expressions of the major hub transcription factors MYC, STAT3, TP63 and KLF4 were verified among LUAD and LUSC, respectively. The upregulated levels of the hub kinases CDK1, GSK3B, CSNK2A1 and MAPK14 were also validated in LUAD and LUSC tumors (*, *p* < 0.001; **, *p* < 0.0001). The Kaplan–Meier curves depict disease-free survival of lung patients with high or low expression in signatures composed of the transcription factors (FOSL2, CHD1, SOX2, KLF4, TP63, STAT3 and MYC) (**c**) or kinases (MAPK14, GSKB3, CDK4, CDK1, MAPK3, CSNK2A1, HIPK2, MAPK8) (**d**) acting as main hubs, respectively. High expression of the signature composed of the hub kinases was significantly associated with disease-free survival (*p* < 0.05), but that of the hubs’ TFs was not. MYC and TP63 were significantly upregulated in LUSC (but not LUAD) patients, exhibited moderate protein expression (**e**,**f**), and showed a reverse pattern of association with patient survival (*p* ≤ 0.05, Log-rank) (**g**,**h**). The TCGA-LUAD and TCGA-LUSC patient cohorts were analyzed using GEPIA2. IHC data were extracted from the Human Protein Atlas.

**Figure 5 ijms-23-10933-f005:**
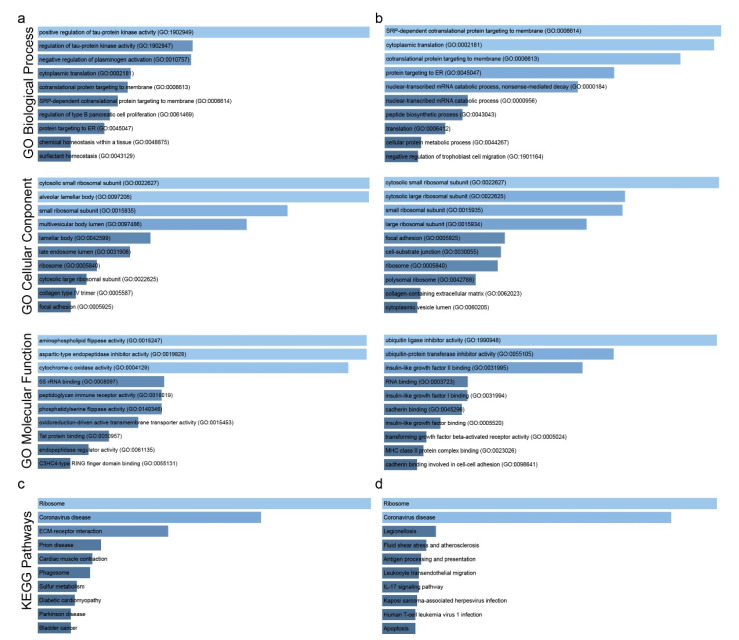
Gene Ontology (GO) enrichment analysis of the co-DEGs in single-gene perturbation experiments in lung cancer. (**a**) GO enrichment results of the co-upregulated genes in lung cancer against the normal tissue. (**b**) GO enrichment results of the co-downregulated genes in lung cancer against the normal tissue. KEGG enrichment analysis regarding the co-DEGs in single-gene perturbation experiments in lung cancer. (**c**) KEGG enrichment for the co-upregulated genes and (**d**) the co-downregulated genes. Bar graphs were sorted using a combined score between the Benjamini–Hochberg (BH)-adjusted *p*-value and the z-score of the deviation from the expected rank. Each bar’s length shows the significance of the corresponding term being relative to the color brightness of each bar.

**Figure 6 ijms-23-10933-f006:**
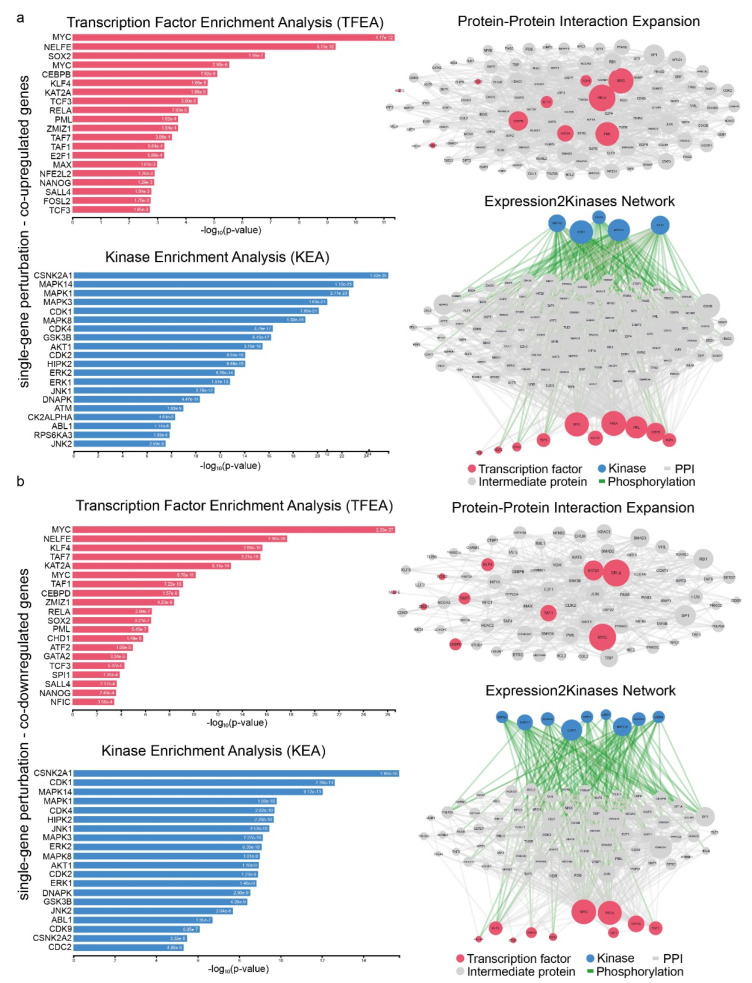
Upstream regulatory networks for co-upregulated (**a**) and co-downregulated (**b**) gene signatures in single-gene perturbation experiments in lung cancer. The networks depict transcription factors (TFs, red nodes), intermediate proteins (gray nodes), and kinases (blue nodes). Gray edges indicate PPI interactions and green edges depict kinase-driven phosphorylation events. Node size is relative to expression. Upstream regulatory networks were constructed using X2K.

**Figure 7 ijms-23-10933-f007:**
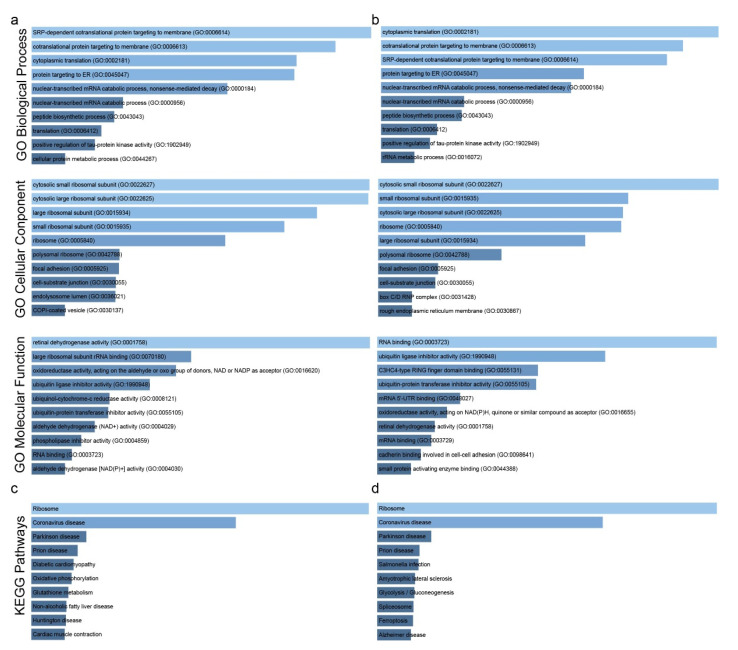
GO enrichment of the co-upregulated (**a**) and the co-downregulated (**b**) genes in drug perturbation experiments in lung cancer. KEGG enrichment for the co-upregulated (**c**) and the co-downregulated (**d**) genes in single-gene perturbation experiments in lung cancer. Bar graphs were sorted using a combined score between the Benjamini–Hochberg (BH)-adjusted *p*-value and the z-score of the deviation from the expected rank. Each bar’s length shows the significance of the corresponding term relative to the color brightness of each bar.

**Figure 8 ijms-23-10933-f008:**
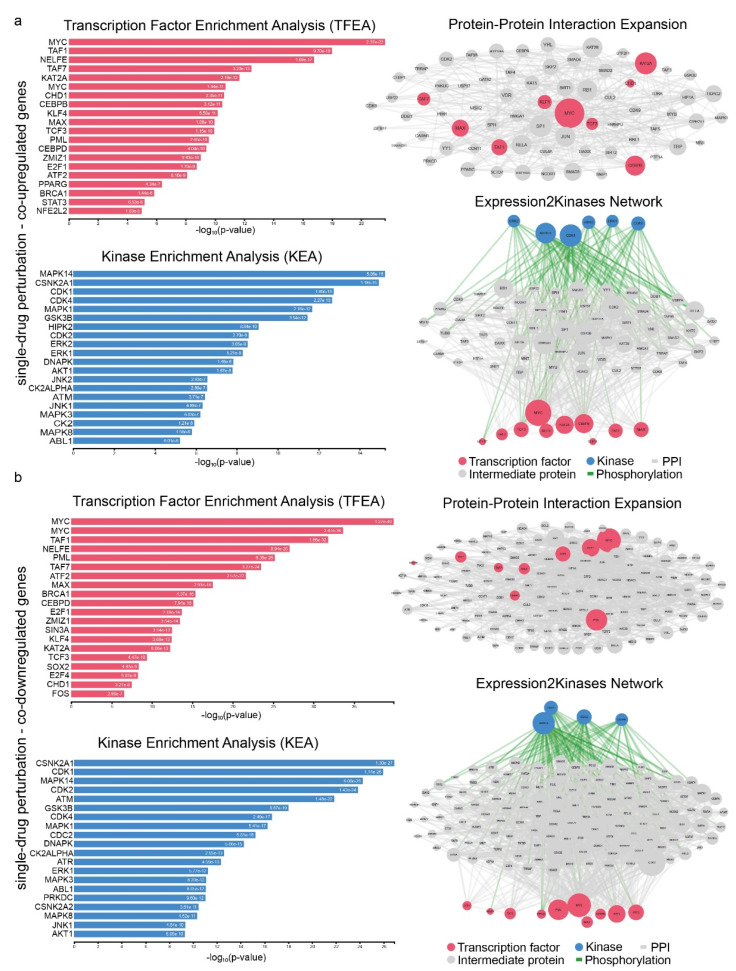
Upstream regulatory networks for co-upregulated (**a**) and co-downregulated (**b**) gene signatures in single-drug perturbation experiments in lung cancer. The networks depict transcription factors (TFs, red nodes), intermediate proteins (gray nodes), and kinases (blue nodes). Gray edges indicate PPI interactions and green edges depict kinase-driven phosphorylation events. Node size is relative to expression.

## Data Availability

The figshare repository was used to store the supporting data of our findings (doi:10.6084/m9.figshare.16736896).

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
