# Peer review of "Signatures of Co-Deregulated Genes and Their Transcriptional Regulators in Lung Cancer"

_ijms, 2022, doi:10.3390/ijms231810933_

Round 1

Reviewer 1 Report (Previous Reviewer 2)

The manuscript is now suitable for publication

Reviewer 2 Report (Previous Reviewer 1)

The authors have addressed my concerns. 

This manuscript is a resubmission of an earlier submission. The following is a list of the peer review reports and author responses from that submission.

Round 1

Reviewer 1 Report

The aim of this study was to decipher the lung cancer gene networks formed by co-deregulated genes (co-DEGs) along with their upstream regulator that were identified using the data retrieved from the Gene Expression Omnibus (GEO). The authors identified critical co-DEGs in lung cancer and provided an insight into their potential use in the development of personalized therapeutic strategies. 

The study was of limited originality in that the data were not generated by the authors but retrieved from GEO. The use of three human lung cancer cell lines to validate the findings of GEO data analyses was questionable because the genomic landscape of lung cancer is far more complicated than that of cultured lung cancer cells. The study would be valuable if the authors were able to validate their findings by providing analysis results based on the data obtained from a new cohort of patients with lung cancer. 

Reviewer 2 Report

The manuscript focuses on in silico analysis of gene expression profile from lung cancer samples in public databases. In my opinion, the authors should improve minor considerations to accept this paper for the publication

- in the text, could the authors define if samples that harbor.drivers gene druggable alterations may represent outloier in this analysis.

- could the authors explain why gep profiles detected on luad cell lines were quite different from in silico analysis?

- could the authors also define how these data may impact on the clinical managment of lung cancer patients

- in the conclusion,could the authors discuss about the absence of real world lung cancet patients population as target to confirm in silico data?